

# Computer simulation on the cueing movements in cue sports: a validation study

Jing Wen Pan[1], Qichang Mei[2,3,4], Justin Fernandez[2,3,5], Hesheng Song[6], John Komar[1] and Pui Wah Kong[1]

[1] Physical Education and Sports Science Academic Group, National Institute of Education, Nanyang Technological University, Singapore
[2] Research Academy of Grand Health, Ningbo University, Ningbo, Zhejiang, China
[3] Auckland Bioengineering Institute, University of Auckland, Auckland, New Zealand
[4] Faculty of Sports Science, Ningbo University, Ningbo, Zhejiang, China
[5] Department of Engineering Science, University of Auckland, Auckland, New Zealand
[6] Department of Physical Education, Guizhou Normal University, Guiyang, Guizhou, China

## ABSTRACT

**Background**. Simulation models have been applied to analyze daily living activities and some sports movements. However, it is unknown whether the current upper extremity musculoskeletal models can be utilized for investigating cue sports movements to generate corresponding kinematic and muscle activation profiles. This study aimed to test the feasibility of applying simulation models to investigate cue sports players' cueing movements with OpenSim. Preliminary muscle forces would be calculated once the model is validated.

**Methods**. A previously customized and validated unimanual upper extremity musculoskeletal model with six degrees of freedom at the scapula, shoulder, elbow, and wrist, as well as muscles was used in this study. Two types of cueing movements were simulated: (1) the back spin shot, and (2) 9-ball break shot. Firstly, kinematic data of the upper extremity joints were collected with a 3D motion capture system. Using the experimental marker trajectories of the back spin shot on 10 male cue sports players, the simulation on the cueing movements was executed. The model was then validated by comparing the model-generated joint angles against the experimental results using statistical parametric mapping (SPM1D) to examine the entire angle-time waveform as well as $t$-tests to compare the discrete variables (*e.g.*, joint range of motion). Secondly, simulation of the break shot was run with the experimental marker trajectories and electromyographic (EMG) data of two male cue sports players as the model inputs. A model-estimated muscle activation calculation was performed accordingly for the upper extremity muscles.

**Results**. The OpenSim-generated joint angles for the back spin shot corresponded well with the experimental results for the elbow, while the model outputs of the shoulder deviated from the experimental data. The discrepancy in shoulder joint angles could be due to the insufficient kinematic inputs for the shoulder joint. In the break shot simulation, the preliminary findings suggested that great shoulder muscle forces could primarily contribute to the forward swing in a break shot. This suggests that strengthening the shoulder muscles may be a viable strategy to improve the break shot performance.

Corresponding author
Pui Wah Kong,
puiwah.kong@nie.edu.sg

**Conclusion**. It is feasible to cater simulation modeling in OpenSim for biomechanical investigations of the upper extremity movements in cue sports. Model outputs can help better understand the contributions of individual muscle forces when performing cueing movements.

## INTRODUCTION

Cue sports are a series of games played with a cue stick on a pool table, including various games, such as 8-ball, 9-ball, 10-ball, snooker, and carom billiards. Among different cue sports games, the cueing movements are largely similar, with multiple practice swings and a final stroke. In the literature, there are only a few studies that have conducted experiments to analyze the biomechanics of the cueing movements. For instance, players' kinematic profiles while executing different types of shots have been reported by a few studies (*Haar, Van Assel & Faisal, 2020*; *Pan et al., 2022*). In addition, a recent case study on one skilled snooker player reported similar biomechanics results when conducting different types of shots, including the stun shot, back spin shot, and top spin shot (*Kong et al., 2021*). In the same study, force platforms were applied to obtain ground reaction forces when the player was performing the cueing movements (*Kong et al., 2021*).

Experimental methods are a conventional means to directly investigate sports movements, and a few studies have been conducted on cue sports to primarily obtain the kinematic variables and ground reaction forces in recent years (*Haar, Van Assel & Faisal, 2020*; *Kong et al., 2021*; *Pan et al., 2022*). However, there are limitations to the traditional experimental methods (*Yeadon, 2004*; *Yeadon & Pain, 2023*). For example, the results are usually sport-specific, and the research outcomes of one particular sport may not be suitably inferred to another. Secondly, while many sports, such as running and jumping, have been well studied, it is difficult to cover all sports given limited number of researchers. The limited research on cue sports is a good example, since there are only a few scientific studies available, most of which solely focused on the ball movements (*Jankunas & Zare, 2014*; *White, 2017*) instead of human movements. Furthermore, regarding human movements, several limitations rendered by traditional experimental methods should be noted. For instance, it is challenging to calculate joint kinetics of the upper extremities, because the forces acting on the cue stick are difficult to quantify. Furthermore, applying traditional experimental methods is unable to identify individual muscle and joint forces, muscle-induced accelerations, and muscle powers. As such, computer simulation could be a useful research tool to compute biomechanical outputs which are difficult to obtain experimentally. In addition, theoretical approaches do not always require collecting experiment data on human participants, but "predict" results based on sound theories, such as Newton's laws of motion. Thus, a combination of experimental methods and theoretical approaches is preferred to address performance-related research (*Yeadon & Challis, 1994*).

In the current literature on cue sports, several attempts have been made to apply computer simulations to conduct investigations. *Szewc, Kudra & Awrejcewicz (2016)* proposed a mathematical model on a circular contact area to present the friction force, moment of the pool balls, and the trajectory of the ball. *Mathavan, Jackson & Parkin (2009)* modeled the balls on a snooker table using video images to determine the ball friction. In snooker, *Legg et al. (2011)* presented novel methods to convert captured videos of balls and table to a 3D model for objective motion analysis of ball movements. These modeling approaches on ball movements facilitated objective evaluations of cue sports performance to supplement subjective observation by coaches and players. However, most studies focused on the balls and there are currently no simulation studies on the player's movements. Hence, it is of interest to use simulation models to investigate human movements and to compute the biomechanical outputs that are difficult to obtain using traditional experimental methods.

Computer simulation models have previously been developed to address research questions on upper extremity movements. In tennis, *Taylor et al. (2009)* applied an upper extremity model to calculate muscle forces during the tennis serve movement. Using a unimanual upper extremity musculoskeletal model, *Seth et al. (2019)* computed the muscle contributions to the scapula and upper arm motions. Thus, using computer simulation methods may also be useful in cue sports to help researchers and practitioners better understand the biomechanics of the cueing movements, in particular for muscle forces that are difficult to determine using experimental methods. Given the advantages of the modeling approach and the factor that no modeling studies have been conducted on cue sports players, the primary purpose of this study was to test the feasibility of applying an existing upper extremity model to investigate the cueing movements. Once the model is validated, further analysis, such as muscle force calculation, could be conducted. The simulation results may help understand the muscle contributions during cueing movements, and facilitate training and skill improvement for cue sports players.

## MATERIALS & METHODS

Two types of cue sports shots in 9-ball were simulated in this study, the back spin shot and break shot. The back spin shot is widely employed in games, with the primary purpose to place the cue ball at a favorable position ready for the next shot. The break shot, the first shot in every frame, is critical as it could influence game dynamics and results (*Pan et al., 2021*). A generic unimanual upper extremity musculoskeletal model was used for both types of shots. The first back spin shot model would be driven by experimental marker trajectories of the upper extremity. The model would be validated by comparing the model-generated joint angle outputs against the experimental results. The second model for the break shot would be driven by marker trajectories and EMG data extracted from the experiment. This model generates model-estimated muscle activation as outputs (*Lu et al., 2020*).

### Participants

The current study was approved by the Nanyang Technological University Institutional Review Board (Protocol Number: IRB-2019-06-037). All participants provided written

consent to participate. The back spin shot was analyzed for 10 male right-handed cue sports players (mean (standard deviation); age 26.7 (8.5) years; height 171.8 (4.7) cm; body mass 66.8 (6.2) kg; playing experience 7.6 (9.1) years). Among the 10 participants, four participants were recruited from the Singapore National Cue Sports team, one was from the university team, and five were recreational players who had roughly 2-year playing experience. The 9-ball break shot was performed by two male cue sports players, who were Participant A (recreational player, age 23 years; height 169 cm; body mass 58 kg; playing experience 5 years) and Participant B (national team player, age 29 years; height 170 cm; body mass 70 kg; playing experience 15 years), both right-handed.

## Simulation model

The generic unimanual upper extremity musculoskeletal model adapted in this present study was previously customized and validated (*Seth et al., 2016*; *Seth et al., 2019*). This model included the torso and right arm, which had six degrees of freedom at the scapula, shoulder, elbow, and wrist. In addition, muscles were included in this model, including trapezius (scapula superior, scapula middle, scapula inferior, clavicle), serratus anterior (superior, middle, inferior), rhomboideus (superior, inferior), levator scapulae, coracobrachialis, deltoideus (anterior, middle, posterior), latissimus dorsi (superior, middle, inferior), pectoralis major (clavicle, thorax middle, thorax inferior), teres major, infraspinatus (superior, inferior), pectoralis minor, teres minor, subscapularis (superior, middle, inferior), supraspinatus (anterior, posterior), triceps (long), and biceps (long, brevis). This model has been applied in several studies on the upper limb movements when executing daily living activities (*Seth et al., 2019*; *Gorkovenko et al., 2020*; *Karimi & Khademi, 2021*; *Fischer et al., 2021*), and is expected to assist assessing functional roles of the muscles in the upper extremity (*Seth et al., 2019*).

## Model customization

The generic model was updated to fit the cueing movements in cue sports. Specifically, the flexion/extension range of motion (ROM) of the elbow joint was set to 155° since full elbow flexion places the proximal forearm against the distal biceps and returns the joint to the outstretched anatomic position with the normal flexion ranging roughly from 150 to 160° (*Lawry et al., 2010*). This is also in line with a previous study on snooker which reported elbow flexion/extension ROM of approximately 145° while performing various cueing movements (*Kong et al., 2021*). Hence, changing from the original 130° to 150° would be more appropriate for the cueing movements. The wrist motions (*i.e.*, flexion/extension and abduction/adduction) were not taken into consideration in the model for simplicity since the wrist joint ROM (*Kong et al., 2021*) and angular velocities (*Haar, Van Assel & Faisal, 2020*) were much smaller than those of the elbow. The mass of a cue stick, which is approximately 550 g, cannot be neglected in the model as it is similar to the mass of one hand (*e.g.*, 70 kg × 0.66% = 462 g) based on the body segment mass reported by *Cheng et al. (2000)*. For simplicity, hand movements were not included in the simulations. Lastly, the pronation/supination motion of the forearm was unlocked and customized for the cueing movements. All simulations of the cueing movements were performed using

OpenSim (version 4.2, Stanford University, Stanford, CA, USA) (*Delp et al., 2007*) on a laptop computer with an Intel i7-10750H, 2.6 GHz processor, and 16 GB of RAM.

Experimental data (marker trajectories during the cueing movements of the back spin shots) were obtained from the experiment sessions. The back spin shot was executed by 10 male cue sports players, and three valid trials were used for analysis for each participant. The testing protocol was adopted from a previous study (*Pan et al., 2022*), whereby the participants were instructed to pot a ball into the middle pocket (Fig. 1A). The cue tip should impact the low part of the cue ball to generate back spin. The participants' upper extremity kinematic data were collected with an eight-camera motion capture system (200 Hz, Vicon MX, Oxford Metrics Ltd., Oxford, UK). To facilitate data collection, 24 markers were fixed on the head, and trunk, upper extremity for each participant, including four markers on the head, four on the trunk, four on the pelvis, seven on the right arm, and five on the left arm. In one cueing movement stroke, the cue stick is first drawn back and then driven forward to hit the cue ball. As such, the stroke can be divided into several sub-phases according to the displacements of the cue stick. All kinematic results (joint angles) were analyzed for the period from the start of forward swing (the cue stick starts to move forward after being drawn back) to the end of the follow through (when the cue stick reaches the farthest point after hitting the cue ball) (*Pan et al., 2022*) using Visual3D (v2021.09.1, C-Motion, Germantown, MD, USA). During this period, the cue stick is driven forward rapidly. The raw marker trajectories were low-pass filtered with a fourth-order Butterworth filter at the cut-off frequency of 10 Hz. Joint angles for the shoulder, elbow, and wrist of the cue-wielding (right) arm were calculated in accordance with the recommendations of International Society of Biomechanics (ISB) (*Wu et al., 2005*; *Gates et al., 2016*). These joint angle data obtained experimentally would be used as reference to validate the subsequent simulation outputs.

The raw trajectories of 11 out of the 24 markers on the upper extremity were exported as the inputs for the simulation model. These 11 markers included RSHO (right acromion process), STRN (suprasternal notch), XIPH (xiphoid process), C7 (7th cervical vertebra), T8 (8th thoracic vertebra), RELM (medial epicondyle of the right elbow), RELL (lateral epicondyle of the right elbow), RWRU (most caudal-medial point on ulnar styloid of the right wrist), RWRR (most caudal-lateral point on radial styloid of the right wrist), R5MC (5th metacarpal head of the right hand), and R3MC (3rd metacarpal head of the right hand). These marker trajectories were low-pass filtered with a fourth-order Butterworth filter at the cut-off frequency of 10 Hz, and then applied to scale the generic model to determine the marker positions on the model (Fig. 2A). Several scale factors were used for scaling the generic model, including 'Humerus' (from RSHO to RELL, from RSHO to RELM; for the scapula and humerus), 'Forearm' (from RELM to RWRU, from RELL to RWRR; for ulna and radius), and 'Hand' (from R3MC to RWRR, from R5MC to RWRU; for hand). The segment masses for the OpenSim model were set according to the results reported by *Cheng et al. (2000)* who applied magnetic resonance imaging (MRI) to provide accurate parameters of Chinese males for simulation modeling. Their results were largely similar to those in the literature (*Dempster, 1955*; *Clauser, McConville & Young, 1969*) with slight differences in the upper arm, thigh, and foot compared with the Caucasian participants

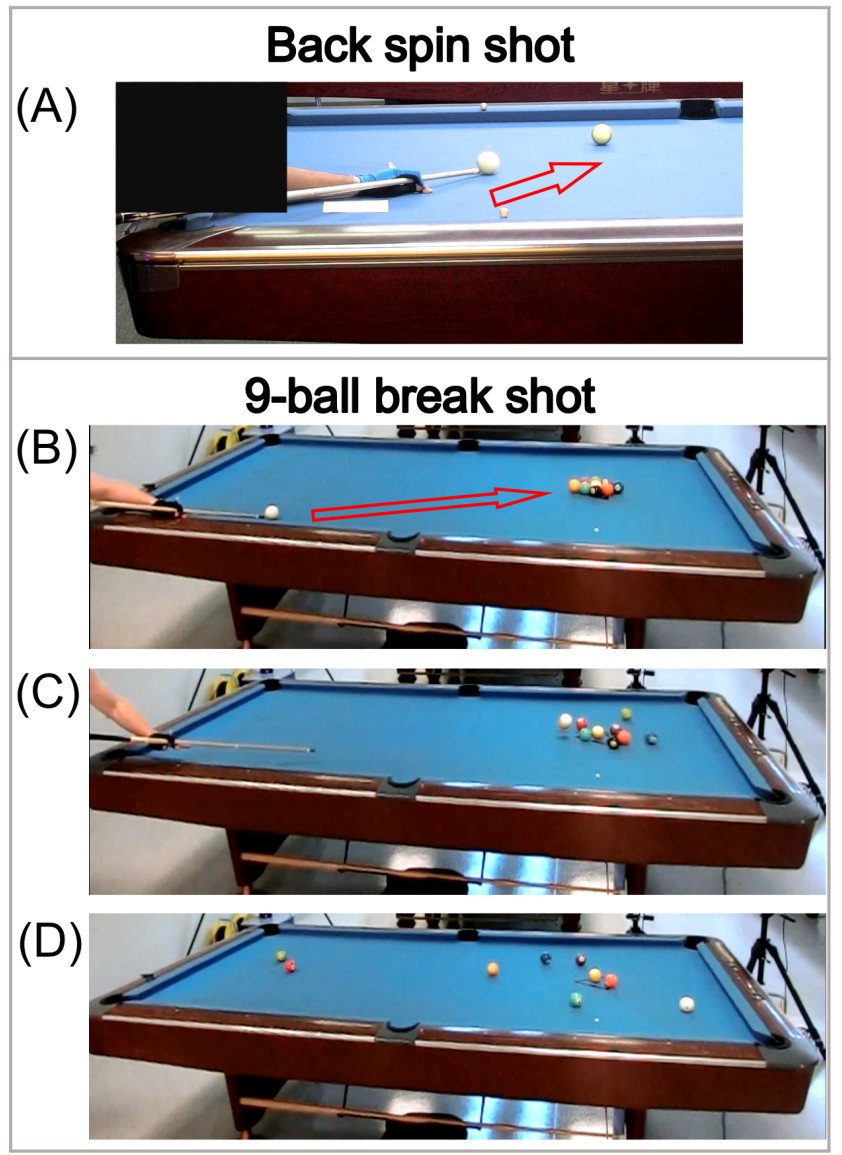

**Figure 1** Testing protocols for (A) back spin shot, and (B–D) 9-ball break shot.

(*Cheng et al., 2000*). Since this present study also recruited male Chinese participants, the results provided by *Cheng et al. (2000)* could be appropriate for scaling the mass of body segment for the participants.

A participant-specific model, fitting each participant's anthropometric dimensions and muscular insertion points, was saved (Fig. 2B). After that, three valid back spin shots were used for simulation, whereby the marker trajectories of each back spin shot were used to drive the model to perform the cueing movements for Inverse Kinematics (Fig. 2C) (*Seth et al., 2019*). In this step, upper extremity joint angles were generated based on the simulation model. The marker trajectories of the simulation model and actual trajectories obtained during experiments were first visually checked. Subsequently, the

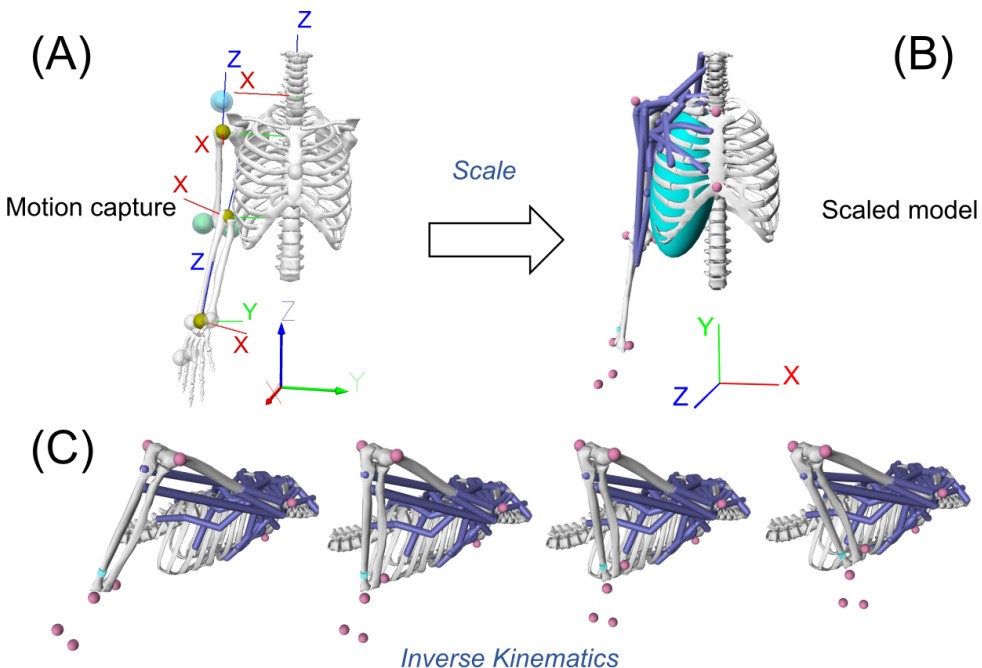

**Figure 2** **Steps of *Scale* and *Inverse Kinematics* on OpenSim.** (A) Experimentally obtained marker trajectories used to scale the model. (B) Scaled model. (C) Step of *Inverse Kinematics* using the marker trajectories of the cueing movement trial to drive the model.

model-generated joint angles were compared against those derived from the experiment, using statistical procedures (details are described under the 'Model validation' section below) (*Mei et al., 2019*).

For the break shot, three valid trials were used for simulation for each of Participants A and B. Besides the marker trajectories during the cueing movements, electromyography (EMG) data were utilized as the simulation model inputs. In each trial, the participants were required to break the racked 9 balls with best effort, in order to make the balls well separated on the pool table (Figs. 1B–1D). The upper extremity kinematic data generated by the simulation model were obtained using the same methods stipulated above for the back spin shot. For Participants A and B, EMG data were collected for five muscles located in the upper extremity of the right side, including trapezius descendens (upper), deltoideus (anterior), deltoideus (posterior), triceps brachii (long head), and biceps brachii (long head). The EMG sensor placement in the present study followed the recommendations of the *SENIAM* project (*Hermens et al., 1999*). After the experiment, the participants were required to conduct maximal voluntary contraction (MVC) trials for the specific muscles, following the recommendations of the *SENIAM* project (*Hermens et al., 1999*), in order to normalize the EMG results (*Ertan et al., 2003*; *Oliver, Plummer & Gascon, 2016*; *Yang & Côté, 2021*). The MVC trials were intentionally conducted after the experiment to avoid possible fatigue effects on the shot performance. The raw EMG data for both MVC and break shot trials were processed with a bandpass filter of 20–450 Hz (*Balshaw et al., 2017*). Then root mean square (RMS) amplitude analysis for EMG signals was conducted using

Visual3D for the same movement period as the kinematic data. At the same time, peak MVC-RMS of each specific muscle was identified, and the break shot EMG-RMS was normalized to peak MVC-RMS.

Model scaling and Inverse Kinematics were performed following the same procedures as those of the back spin shot. After that, Computed Muscle Control (CMC) was executed to obtain EMG-driven simulations with both kinematic and EMG inputs (*Thelen, Anderson & Delp, 2003*; *Seth et al., 2019*). For this simulation, kinematic inputs were joint angles generated in the Inverse Kinematics step and the EMG inputs were the processed experimental data. In this step, muscle excitations (*e.g.*, muscle activation level, muscle force, muscle fiber length) were computed as model outputs. For the EMG inputs, 0% was considered not activated for the particular muscle, and 100% as fully activated (*Mei et al., 2019*; *Lu et al., 2020*). A model-estimated muscle activation was performed accordingly (*Lu et al., 2020*). The muscle activation timing ranged from 0 to 100% of the cueing movement in a break shot. Muscle forces were normalized to the individual body weight (BW) (*Demers, Pal & Delp, 2014*; *Hall et al., 2019*), and expressed as BW%.

## Model validation

To validate the model for the back spin shot, two types of comparisons were made between the upper extremity joint angles calculated using visual3D (experimental data) and those generated using the OpenSim model. Firstly, statistical parametric mapping (SPM1D) (*Nichols & Holmes, 2002*; *Pataky, 2012*) was applied to compare the entire waveforms of the angle-time waveforms of shoulder elevation, elbow flexion/extension, and elbow pronation/supination between the two sets of data using MATLAB (version 2021b, MathWorks, Natick, MA, USA) (*Mei et al., 2019*), whereby all experimental and OpenSim data were normalized to 101 data points using a liner interpolation method. Secondly, to examine the discrete variables which have been commonly used in biomechanics studies, joint ROM were compared between the two sets of data. Specifically, shoulder elevation, elbow flexion/extension, and elbow pronation/supination obtained using OpenSim and experiments were compared with paired-samples t test using the JASP statistical software (version 0.17.1; *JASP Team, 2023*), whereby the data normal distribution was confirmed by the Shapiro–Wilk test. All statistical significance was set at the 0.05 level.

Since only two participants were selected for the simulation of the break shot, no inferential statistical analysis was conducted. Instead, the muscle activation data (EMG) obtained from the experiment and simulation model were plotted to facilitate model validation.

## RESULTS

### Back spin shot

To visually check the quality of the simulation, marker trajectories during the cueing movements were compared between the experimental data (blue markers) and model-generated data (pink markers) (Fig. 3). To facilitate this comparison, after Inverse Kinematics, the marker trajectories of the cueing movement were associated with the OpenSim model to show both experimental and model-generated markers. As shown

(A)
(B)

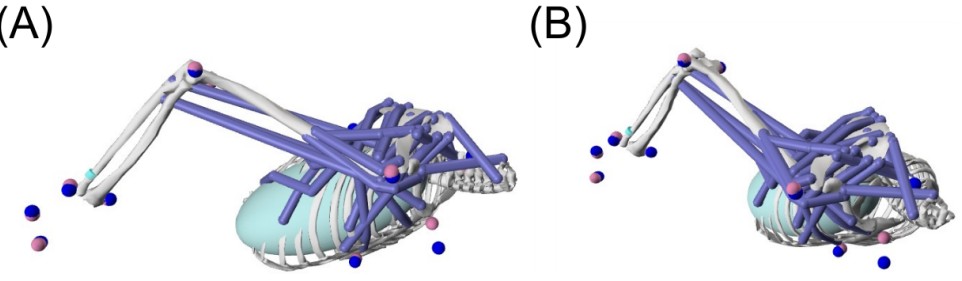

**Figure 3  Visual comparison of the marker trajectories between the OpenSim-generated maker data (pink) and experimental marker data (blue) from (A) the side view and (B) front view.**

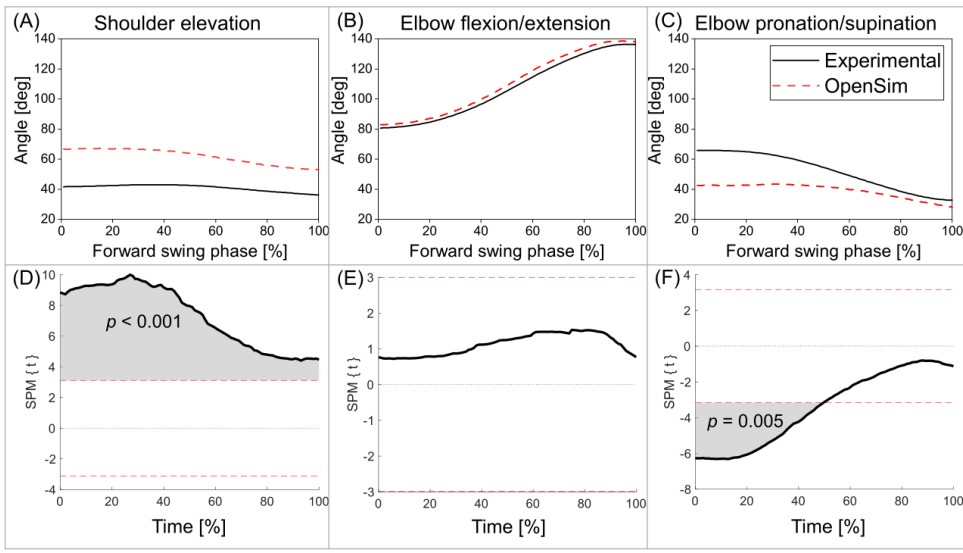

**Figure 4  Comparisons of the OpenSim-generated and experimental joint angles.** Joint angles of (A) shoulder elevation, (B) elbow flexion/extension, and (C) elbow pronation/supination of the OpenSim-generated and experimental data. Result of statistical parametric mapping (SPM1D) of the entire wave-forms for the angle-time curves of (D) shoulder elevation, (E) elbow flexion/extension, and (F) elbow pronation/supination between the two sets of data.

in Fig. 3, the OpenSim markers and experiment markers corresponded well on the cue-wielding (right) arm, while visible discrepancies were found for the trunk markers.

In terms of the kinematic comparisons, the results of SPM1D showed that the entire angle-time waveforms of the shoulder elevation angles significantly differed between the OpenSim-generated and experimental data (Fig. 4D, from 0 to 100%, $p < 0.001$), although the two waveforms showed similar tendency (Fig. 4A). No significant difference was found throughout the entire waveforms of the elbow flexion/extension angles (Fig. 4E, $p > 0.05$) between the two sets of data. For the elbow pronation/supination angles, there was a significant difference in the first half of the cueing movement (Fig. 4F, from 0 to 50%, $p = 0.005$).

**Table 1  Comparisons of the joint ranges of motion (degrees) between the OpenSim-generated and experimental results.**

|  | OpenSim | Experimental | Mean diff | 95% CI | $p$ | Effect size | |
|---|---|---|---|---|---|---|---|
| Shoulder | 16.3 (12.4) | 9.1 (4.4) | 7.2 | [0.3, 14.1] | **0.042*** | 0.749 | Medium |
| Elbow F/E | 56.7 (13.9) | 56.3 (13.0) | 0.4 | [−3.7, 4.5] | 0.827 | 0.071 | Trivial |
| Elbow P/S | 19.8 (10.3) | 33.9 (14.2) | −14.1 | [−22.5, −5.7] | **0.004*** | 1.206 | Large |

Notes.

Data are expressed as mean (standard deviation). Shoulder denotes the range of motion of shoulder elevation. Elbow F/E denotes the range of motion of elbow flexion/extension; Elbow P/S denotes the range of motion of elbow pronation/supination. Mean diff denotes the mean difference between the OpenSim-generated kinematic and experimental data (Mean diff = OpenSim − Experimental). CI denote confidence intervals.

Significant difference ($p < 0.05$) is shown in bold text and indicated by an asterisk.

The results of t test showed no significant difference in the elbow flexion/extension ROM between the OpenSim-generated and experimental data (Table 1, $p = 0.827$). For the shoulder elevation ROM, the OpenSim-generated data were significantly greater than the experimental results ($p = 0.042$, medium effect size). The elbow pronation/supination ROM was significantly smaller in the OpenSim-generated data than the experimental results ($p = 0.004$, large effect size).

## Break shot

The OpenSim-generated muscle activations were visually compared against the experimental EMG data (Fig. 5) for Participants A and B. The model outputs did not show a good agreement in the shoulder muscle activations between the two sets of data, in particular for the trapezius of Participant B and deltoideus (anterior) of Participant A. For the deltoideus (posterior), triceps brachii, and biceps brachii, the time-varying waveforms corresponded well between the two sets of data. Muscle forces (Fig. 6) were extracted for the deltoideus (anterior), deltoideus (posterior), triceps brachii (long head), biceps brachii (long head), and biceps brachii (short head). The force of the trapezius was not included due to the relatively poor simulations of this specific muscle found in the previous steps.

## DISCUSSION

This study tested the feasibility of implementing computer simulation approach to investigate cue sports players' cueing movements. Two types of shots, namely the back spin shot and 9-ball break shot, were selected for simulation. The cueing movements in both types of shots are not identical but similar, with the shared sub-phases in a stroke. For example, during back swing, the cue stick is drawn backward, and then driven forward rapidly during the forward swing phase. The feasibility of using the generic unimanual upper extremity musculoskeletal model (*Seth et al., 2019*) to investigate cue sports movements was verified as reflected by the generally good model validation results. After the model was validated, it was applied to calculate muscle forces for the upper extremity muscles in the 9-ball break shot.

### Back spin shot (model validation)

This present study selected three joint angles for analysis, including shoulder elevation, elbow flexion/extension, and elbow pronation/supination angles, all of which are greatly

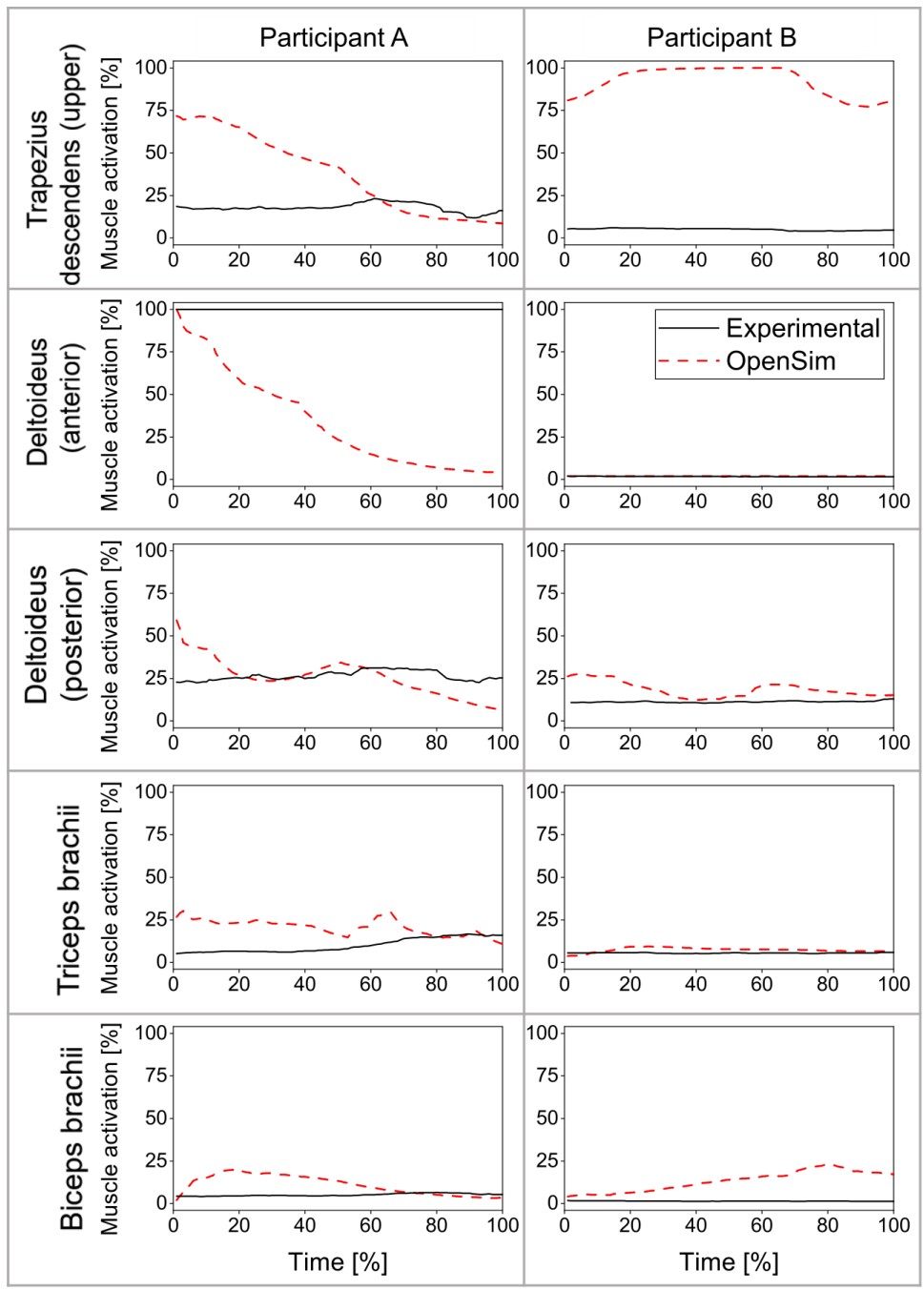

**Figure 5** Comparisons of the upper extremity muscle activations between the OpenSim-generated and experimental data, wherein the EMG-RMS was normalized to the MVC-RMS.

engaged in the cueing movements. The results showed a significant difference in the shoulder elevation angles (Fig. 4, from 0 to 100%, $p < 0.001$) between the OpenSim-generated and experimental data although these two waveforms showed a similar tendency. Several factors likely contributed to the discrepancy in the shoulder elevation angles between

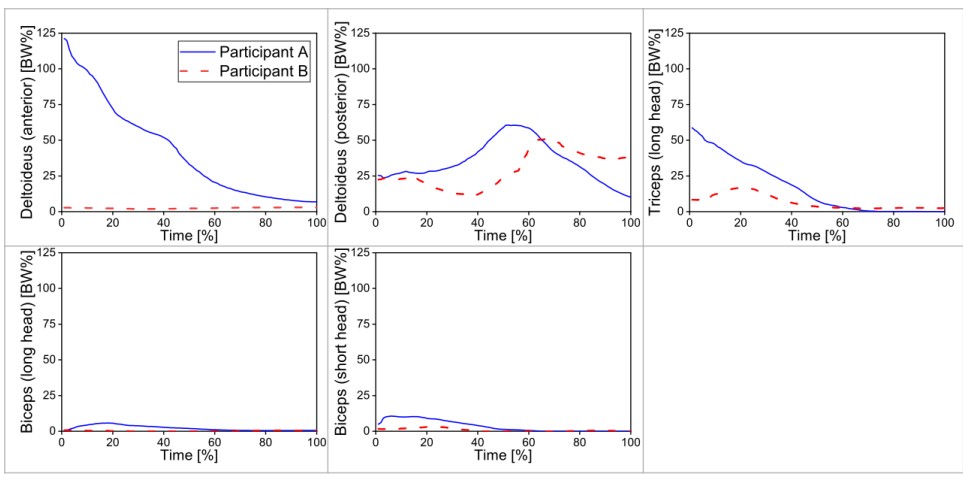

**Figure 6 Muscle forces of the cue-wielding arm muscle of two participants when performing the break shots.** All muscle forces were normalized to the individual body weight [BW%].

the two sets of data. Firstly, in the work by *Seth et al. (2019)*, a few sensors were placed on the thorax, scapula, and humerus to obtain the kinematics of the upper extremity joints, in particular for the shoulder. However, in the current study on the cueing movements, no markers or sensors were fixed on the scapula, while markers were only put on the acromion (RSHO) and spine (C7, T8). The scapula motions could deviate from the experimental results and led to incorrect shoulder elevation angles in the simulations. In addition, the model scaling in OpenSim may not be anatomically accurate by simply scaling the bone length according to the marker data while neglecting the possibly varied bone shapes. A statistical shape modeling could be plausible to address this issue (*Zhang et al., 2016*). Hence, future studies should include more detailed shoulder data (*e.g.*, scapula motion) and better scaling methods to improve the simulations, aiming to reduce the discrepancy between experimental and OpenSim-generated biomechanics results.

For the elbow, the OpenSim-generated joint angles corresponded well with the experimental data. For the elbow flexion/extension angles, no significant difference was found in the entire angle-time waveforms between the two sets of data. However, for the elbow pronation/supination angles, a significant difference was observed for the entire waveforms from 0 to 50%. In addition, greater elbow pronation/supination ROM was found in the OpenSim-generated results than the experimental data according to the results of the t test (Table 1, $p = 0.042$). The elbow pronation/supination angles represent the rotations of the forearm during the cueing movements. While the makers fixed on the anatomical landmarks (*e.g.*, RSHO, RELM, RELL, RWRU, RWRR) were sufficient for the joint angle calculation on Visual3D, the lack of tracking markers on the forearm during the cueing movements may lead to the relatively poor kinematic data generated using the OpenSim simulation models. Future studies should consider adding a few tracking markers (*e.g.*, marker cluster) on the forearm to obtain better kinematic data for simulating the elbow pronation/supination motions.

Collectively, the generic unimanual upper extremity musculoskeletal model (*Seth et al., 2019*) together with the experimental kinematic data could run good simulations of the elbow motion but not yet shoulder motion. Hence, it is feasible to customize the generic unimanual upper extremity musculoskeletal model (*Seth et al., 2019*) to cater for biomechanical investigations of the upper extremity movements in cue sports. Further studies are needed to refine the model with more detailed data (*e.g.*, scapula motion, forearm motion) as model inputs to improve the simulation of shoulder motion.

## Break shot (model validation and muscle force calculation)

The CMC step utilizes the marker trajectories as well as the EMG data as the model inputs, and estimates the muscle force variables that are difficult to measure directly from experimental methods alone (*Seth et al., 2019*). After executing the CMC step, muscle activations were extracted for the upper extremity muscles by the simulation model to validate against the experimental EMG data. The results of the break shot simulation showed that OpenSim-generated shoulder muscle activations deviated substantially from the experimental data in the two participants (Fig. 5). The mismatch could also be attributed to the relatively poor simulation for the shoulder joint owing to the lack of sufficient scapula motions. For triceps brachii and biceps brachii, the time-varying waveforms were similar between the two sets of data. This suggests good simulations for the upper arm muscles.

After validating the simulation model, muscle forces were calculated for the two participants. As the cueing movement was analyzed for the period from the start of forward swing to the end of follow through, the early phase in the entire waveform should be the forward swing phase, and the latter should be the follow through phase. For both participants, according to the preliminary results, the biceps brachaii did not contribute much to the forward swing (early phase) while the deltoideus forces were great during the forward swing phase (Fig. 6). Great triceps brachii force may contribute to the shoulder movement and stabilize the shoulder joint during forward swing. The current results indicated that the cueing movement may primarily rely on the shoulder muscles instead of the biceps brachii. Players are hence recommended to focus on training the triceps brachii and shoulder muscles, which could generate a high cue tip speed. The deltoideus anterior force of Participant A (>100% BW, Fig. 6) was much greater than that of Participant B. This great discrepancy could be due to the different muscle activation patterns between the two participants. More simulations on different participants are needed to evaluate the muscle activations across individuals to confirm the current preliminary results. Additionally, since some players may have overuse injuries in their neck or shoulder, it is of importance to investigate the impacts on these parts and/or inappropriate muscle activations. Using experimental methods alone, it is difficult to directly measure the upper extremity kinetics and muscle forces during cueing movements. Using simulation models can help obtain these important kinetic variables. Therefore, simulation models are promising tools to help better understand the biomechanics and injury risks associated with cue sports.

## LIMITATIONS

There were a few limitations to the current study. Firstly, the simulations for the shoulder joint were poor, which was primarily due to the lack of the marker trajectories on the shoulder joint. Future studies should focus on the optimizations of the shoulder (*e.g.*, by obtaining detailed scapula motions). At the same time, other model scaling methods, such as the statistical shape modeling (*Zhang et al., 2016*), could be considered to improve the model quality. Secondly, the relatively poor simulation for the elbow pronation/supination could be owing to the lack of tracking markers fixed on the forearm. When collecting experimental data for simulations, future studies should consider including a few markers (*e.g.*, marker cluster) on the forearm to provide more kinematic data as model inputs. Lastly, the sample size was small in this study owing to the constrains in time and manpower, especially for the break shot simulation which required substantial effort to construct the model. Once the model is further refined, further work can be done to run simulations for more participants. In-depth analysis of muscle forces can also be conducted to explore the relationship between muscle activations and shot performance.

## CONCLUSIONS

This study demonstrated that it is feasible to customize the generic unimanual upper extremity musculoskeletal model (*Seth et al., 2019*) on OpenSim for biomechanical investigations of the upper extremity movements in cue sports. While the elbow motion matched well between the simulation outputs and experimental results, the OpenSim-generated shoulder kinematics and muscle activations deviated from the experimental data. Based on the preliminary results, the estimated muscles forces were pronounced for the shoulder joint during the 9-ball break shot. Players should acknowledge that shoulder muscles play an important role in the break shot. Strength and conditioning training targeting shoulder muscles may be considered to enhance the break shot performance. Future studies can verify the present findings with a larger sample size.

## ACKNOWLEDGEMENTS

The authors would like to thank Mr Michael Chan for his assistance in data collection. We gratefully acknowledge Ms Dawn Guo and Ms Mui Kheng Tay for their help with the experiment pool table arrangement. Appreciation is expressed to Ms Lily Liew and Mr Sharik Sayed from Cuesports Singapore Academy.

### Funding

This project was funded by the National Institute of Education Academic Research Fund (NIE AcRF, RI 1/19 KPW). The first author (JWP) of this article was supported by the China Scholarship Council (CSC). The funders had no role in study design, data collection and analysis, decision to publish, or preparation of the manuscript.

## Grant Disclosures

The following grant information was disclosed by the authors:
National Institute of Education Academic Research Fund: NIE AcRF, RI 1/19 KPW.
China Scholarship Council (CSC).

## Competing Interests

The authors declare there are no competing interests.

## Author Contributions

- Jing Wen Pan conceived and designed the experiments, performed the experiments, analyzed the data, prepared figures and/or tables, authored or reviewed drafts of the article, and approved the final draft.
- Qichang Mei conceived and designed the experiments, analyzed the data, prepared figures and/or tables, authored or reviewed drafts of the article, and approved the final draft.
- Justin Fernandez analyzed the data, authored or reviewed drafts of the article, and approved the final draft.
- Hesheng Song analyzed the data, authored or reviewed drafts of the article, and approved the final draft.
- John Komar conceived and designed the experiments, authored or reviewed drafts of the article, and approved the final draft.
- Pui Wah Kong conceived and designed the experiments, authored or reviewed drafts of the article, and approved the final draft.

## Human Ethics

The following information was supplied relating to ethical approvals (*i.e.*, approving body and any reference numbers):

The current study was approved by the Nanyang Technological University Institutional Review Board (Protocol Number: IRB-2019-06-037).

## Data Availability

The data are available at the NIE Data Repository: KONG, Pui Wah, 2023, "Computer simulation on the cueing movements in cue sports: a validation study", https://doi.org/10.25340/R4/OCDJTG, NIE Data Repository, V1, UNF:6:E4hsFqB1gpOJ40Qih1j2Yg==[fileUNF].

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
