# Peer review of "Computer simulation on the cueing movements in cue sports: a validation study"

_PeerJ, doi:10.7717/peerj.16180_

## Round 0.1 · original submission · Major Revisions

Both reviewers believe the research has been completed with rigour and to a high technical standard.

However, they struggle to understand the research question and aim put forward. Kindly address the reviewers´ concerns point-by-point.

·

Basic reporting

The paper describes a pilot study to simulate two kind of movents made by skilled subjects performing cue sport. The language is clear and the paper is well structured. The experimental design is quite clear, but some improvements can be done to better describe the protocol.
0) the main objective of the paper is not clear: is the experiment useful to verify whether the OpenSim Project can be used to simulate movements in cue sports or to highlight the differences between skilled subjects to optimize the performance or both?
1) the authors studied two kind of shots: a figure (better a video, if possible) to visualize whch kind of movements are can better explain what are they working on, since no all the reades are expert in cueing. At line 178, static versus dynamic trials for the same back spin shot are cited... is there a difference?
2) Were the shots operator guided (same starting position, same movements length, etc) or did each subject perform the movement as in his personal past experience? I think that it is easy to perform similar shots with completely different shoulders and arms positions and movements. In this case, the differences obtained for some kind of movemnents and reported in the paper are fully justified (expecially for the last comparison between the 2 subjects). Probably, a drawing of starting positions of shoulders and arms (if fixed) can be useful.
3) LIne 109-111: it is not clear how the validation was made: do you use the same values both for model inputs and then verification?
4) All the abbreviations in the article should be explained (STRN, XIPH, RELL, etc). Please verify all the text.
5) at line 118 and following; is the number in brackets the standard deviation? Please specify the meaning of this value.
6) you have used parametric analysis (paired t test)... please justify this choice.
7) the insertion of a grid in figures 3, 4 and 5 can be useful to highlight the graphs.
8) Avoid repetitions in the abstract. "Results. The results showed..." or "Conclusions. In conclusion...". If the title of the paragpraph is "Conclusion" you have not to state the same concept it in the following proposition.

Experimental design

see below

Validity of the findings

see below

Additional comments

see below

Reviewer 2 ·

Basic reporting

The manuscript uses professional English throughout, however, the structure of the manuscript especially within the introduction, methods, and discussion sections which make it difficult to understand the rationale for the research aim and follow the processes undertaken.

Specific comments:
1. The introduction currently introduces the benefits of the research approach (computer simulation methods) and a review of the previous literature on cue sports before concluding that using computer simulation methods might be useful to investigate cue sports.

Firstly, there is no description or explanation of what cue sports are or what the cue movement is, which is not a huge issue if you have prior knowledge of the sport, but given this paper is currently structured to focussing on computer simulation and whether it can be used to investigate an upper limb movement, this is not guaranteed. In addition, this study focussed on two different ‘shots’ produced by the cueing movement, with no explanation of what these are either.

Secondly, the concluding statement ‘using computer simulation models is useful to help researchers and practitioners better understand the biomechanics of sporting movements as well as finding the optimal parameters of the model leading to enhanced sport performance’ is true but already known. It is very generic and not specific to cue sports, and is not a strong rationale for the research aim.

I recommend that the introduction is rethought and rewritten to focus on cue sports and how computer simulation could be used as a tool to answer the current gaps in the literature, rather than focussing on computer simulation and whether it can be used to investigate cue sports. In addition, some literature on upper limb models should be introduced in the introduction- currently this is with then methods (L149-154).

2. The materials and methods section is split into four sections which do not intuitively flow and leads to repetition when explaining the process for the two shot types. An initial overview of the participants and the modelling process is provided prior to sections on the experiments, computer simulations, and model validation. Whilst I appreciate the difficulty in writing a methods section for a theoretical study which includes experimental data concisely so that information is not introduced before it is explained, the current flow requires revision to improve the numerous examples of repetition (e.g. L108 & L117; L122-128 & L172-L180).

I recommend utilising the following sub-headings (with the following information)
- Participants: include all information on the participants in one place
- Simulation model: describe the simulation model(s) in detail (more information is required on its structure than is currently provided in L149-154), and what inputs are required
- Model customisation: describe how the model was customised and evaluated using experimental inputs.
- Model validation – describe how the model was validated for each shot type (which is how this section is currently structured.
Within the methods section, more clarity is required to highlight that the process were different for the two different shots. In the results, you used sub-headings – it might be worth introducing these within the methods were appropriate.
3. The discussion again starts with a generic overview which reads like a conclusion. This information is then discussed in more detail in the specific sections for the backspin and break-shots. I recommend removing this or removing the elements about the specific findings, which are discussed further on.
4. The figures are relevant, high quality, well labelled, and described – only minor comment if relevant in the digital world is that I printed the manuscript in Black and White and it was not possible to distinguish blue and pink (Figure 2) although the difference in Figures 3 & 4 were distinguishable. Will leave this with the authors to decide if this needs to be changed.
5. The data appears to be available via the NIE data repository.

Experimental design

1. As previously stated (Basic Reporting point 1) the submission does not clearly define the research question. In the introduction the aim is stated as ‘this study would attempt to test the feasibility of applying simulations models in cue sports to investigate cueing movements’ whilst the discussion opens with ‘This study attempted to implement a computer simulation approach to investigate cue sports players cueing movement’.

In reality, this manuscript investigates the validity of a computer simulation model to recreate the cueing movements utilised in cue sports. Whilst the authors make it clear that there is a lack of research in cue sports, it is not entirely obvious that a valid model will enable future investigation of the cause and effect relationships which exist in these movements.

2. The investigation has been completed with rigour and to a high technical standard, as well as in line with the ethical standards in the field.

3. The methods provide enough detail to reproduce this study, however, as previously stated the conciseness needs improving, as well as the model detail which is currently provided by directing the reader to a previous literature on an upper limb model.

Validity of the findings

1. The data is robust, statistically sound, and controlled but at times it is unclear how this was simulated (see additional comments). It has also been made available.

2. The conclusions are appropriately stated at the end of the discussion, however, these are then used to discuss the differences between the different types of shots which was not the purpose of this paper, and is not appropriate based on the lack of statistical inference, and participant numbers. This likely needs revisiting once the research aim is confirmed.

Additional comments

Abstract
L45 – this is not background. Although it is an aim which is closer aligned to the manuscript than those within it.
L47 – more detail is required on the model – how many segments, what type, whose etc.
L51 – how was the experimental data collected, what was it.
L51-61 – add some statistical details in to validate the claims.
Methods and Materials
L118 – what level are these participants?
L120 – explain what the testing protocol is and then reference the previous study.
L123 – what is the forward swing and the end of the follow through? How are these defined?
L149 – describe the model and reference it’s origin. How many segments, degrees of freedom, muscles etc. It makes little sense when things are locked or removed if they were not described initially.
L172 – the section describing how the kinematic data was determined for a second time is not required – could write this section uniformly for both shots.
Results
L225-L228 – this should be in the methods.
L232 - did you normalise your joint angles so they were comparable? For example, did the anatomical position in both models correspond to the same kinematic values?
L232 – did you complete any dynamic functional calibrations to explore the effect of soft tissue movement affecting the experimental data verse the simulated data? Could the differences at certain points be related to soft tissue movement and marker placement when the arm is more or less flexed?
L232-234 – if the kinematics have been input to drive the model for the back-spin shots why are there differences between the experimental and simulated kinematics?
Discussion
L279-283 – should the justification of using SPM1D be the headline of this section?
L330 – this should potentially be in the methods. How were the muscle parameters scaled/defined for each individual?

---

## Round 0.2 · Major Revisions

One reviewer suggests providing further elaboration on the modeling procedures for readers unfamiliar with OpenSim.

·

Basic reporting

The authors have corrected the paper accordingly with my previous suggestions and they answered to all my questions. The changes introduced in the text are clear and make the paper suitable for publications.

Experimental design

no comment

Validity of the findings

no comment

Additional comments

no comment

Reviewer 2 ·

Basic reporting

Thank you for updating the manuscript and taking on board the feedback of both reviewers.

I have a few further comments to address before I am satisfied that the paper is ready for acceptance.

Specific comments:
1. Thank you for adding the section explaining what cue sports are. The structure of the introduction however could still be improved.

The current flow with the paragraphs is as follows:
a. What is wrong with experimental methods.
b. Some theoretical research on cue sports focussing the movement of the ball
c. Some literature on experimental research on cue motion and upper body simulation models finishing with the aim of validating an upper arm model to investigate cueing movements.
While I believe that this is a suitable research aim, I do not think the introduction makes a very clear argument for this research aim in its current format.

A potential solution could be to…
a. Explain what cue sports are and what is currently known and unknown
b. Explain why experimental methods are limited in answering the research question and that a theoretical approach might be required/better
c. Explain what has been done theoretically in cue sports and in other sports around the upper limb
d. Propose the aim of the research to validate an upper limb model to investigate cue motion

I would also recommend considering changing the title from “A pilot study” to “A validation study”
Finally, this section would also benefit being checked from an English grammar perspective.

2. Thank you for updating the methods and materials.

a. Personally I think the section added between L125 and L136 is not necessary as most of the information is in the later sections.

b. The information added before L183 is helpful and improves the understanding of the model.

Unfortunately, the section from L184 to L222 is confusing, and I do not understand how the experimental data was used to drive the model. You describe the experimental protocol to collect the data, but do not provide any information on the marker set or positions of these markers. You explain that you are interested in the phase between the start of the forward/ swing and follow through and this was analysed in Visual3D with the raw marker trajectories filtered at 10Hz and joint angles calculated.

In the next paragraph, there is information regarding the markers required for the model, I assume these markers match those used in the experimental data but this is not obvious. You then state the markers were filtered again in Nexus and this matched the visual3d cut-off frequency. At this point, I am unclear what the two sets of different markers are and how they were collected. This paragraph then moves onto describe the inertia inputs for the model.

My question therefore, is there two sets of inputs? If so explain what the two sets of data are more clearly, or is this an accidental repetition, or have you filtered the data twice?

c. The next paragraph discussed how the model was further scaled – I assume this is step a to b in figure 2 which is defined from the static trials (I am unsure why you have removed the word static here). Maybe you could refer to Figure a and b in the text.

In addition, it is not clear what ‘inverse dynamics’ means especially for those not used to using OpenSim– I think you have driven the model using the marker trajectories (as this is mentioned below as inputs) and the joint angles were output by the model? It might be helpful to describe what the inputs were driving the model and what the outputs were that were used for validation.

d. On L252 you refer to CMC – it might be helpful to explain what this is for those unfamiliar with OpenSim.
3. The model validation section reads well and is clear how the model was validated. However, in the results you discuss that you visually checked the marker trajectories between the experimental data and the model generated data. I have a couple of issues with this:
a. If this is part of the model validation it needs describing in the methods
b. Intuitively if you are driving the model using the marker trajectories surely these should match? I guess there is some element of fitting that goes on with OpenSim to ensure the segment lengths remain the same length – is this part of the modelling process or part of the different ways of utilising markers to determine joint angles in rigid body systems? I would be tempted to remove this section and start the results at Line 290.
4. Thank you for making the changes to the discussion it reads much better.
5. Thank you for making the changes to the figures.

Experimental design

1. The experimental design is appropriate and the investigation has been completed with rigour and to a high technical standard, as well as in line with the ethical standards in the field. However, as explained above the methods requires more detail to clarify how this study was completed.

Validity of the findings

1. The data is robust, statistically sound, and controlled but at times it is unclear how this was simulated (see general comments). It has also been made available.

2. The conclusions are appropriately stated at the end of the discussion.

Additional comments

Abstract
L45 – consider rewording ‘Computer simulations models have previously been applied to analyse sporting movements of the upper extremity. However, it is unknown if these musculoskeletal models are suitable to investigate cue sports movements. This study aimed to investigate the validity of using an upper extremity musculoskeletal simulation model to analyse kinematics and predict muscle forces in cue sports movements.’
L50 – consider explaining the model before the data used to validate it
Introduction
L117 to L122 – consider rewording this section to be more concise regarding the aim rather than what will be done if it is valid.
Methods and Materials
L149 – Delete the last sentence
L235 – change separate to separated
Results
L306 – Delete first sentence and start at ‘The model generated muscle activations….’
Discussion
L325 – delete – in this study
L331 to L335 – consider deleting it does not offer anything to the discussion.

L350 – what do you mean by better simulations?

---

## Round 0.3 · Minor Revisions

Please address the minor comments by the reviewer.

Reviewer 2 ·

Basic reporting

Thank you for updating the manuscript and taking on board my feedback.

Based on my previous review:

1. Thank you for updating the title and introduction it reads much better especially the English and makes a much stronger argument for this manuscript.

2. Thank you for updating the methods and materials.

a. Thank you for highlighting this section (L124-143) was previously requested by the other reviewer. From my perspective, I think the part starting on L137 onwards is not required – but it is not a major issue, and I am happy if it remains as currently submitted.

b. Thank you for editing the section to clarify how the markers were used. This section now is a lot clearer to understand.

c. Thank you for editing this section to clarify what the inputs and outputs of the inverse kinematic process were. This section now reads much clearer.

d. Thanks for editing the CMC to describe what these outputs were.
3.
a. Thank you for adding the visual check into the methods.
b. Thank you for providing a justification regarding the marker trajectory comparison. I agree with your explanation and understand the methodology. My concern was that the manuscript previously read - we have input 11 markers to drive the model, and we are then going to compare those markers visually to see how well they match.

Experimental design

1. The experimental design is appropriate, and the investigation has been completed with rigour and to a high technical standard, as well as in line with the ethical standards in the field. The method has been updated to provide more detail to clarify how this study was completed.

Validity of the findings

1. The data is robust, statistically sound, and controlled. It is also clear how this was simulated. It has also been made available.

2. The conclusions are appropriately stated at the end of the discussion.

Additional comments

Please consider these minor comments...

Introduction
L92 – Consider changing to ‘Experimental methods are a conventional means to directly investigate sports….’
L119 – Consider changing to ‘Computer simulation models have previously been developed to address research questions on upper extremity movements.’
L129 -Change ‘would’ to ‘could’
Methods
L276 – Consider changing ‘back spin test’ to ‘back spin shot’

---

## Round 0.4 · accepted · Accept

The reviewer is happy with this version of your manuscript.

Reviewer 2 ·

Basic reporting

I am happy that this manuscript is suitable for acceptance. Well done all.

Experimental design

No Comment

Validity of the findings

No Comment